# Biological Activities and Chemical Profiles of *Kalanchoe fedtschenkoi* Extracts

**DOI:** 10.3390/plants12101943

**Published:** 2023-05-10

**Authors:** Jorge L. Mejía-Méndez, Horacio Bach, Ana C. Lorenzo-Leal, Diego E. Navarro-López, Edgar R. López-Mena, Luis Ricardo Hernández, Eugenio Sánchez-Arreola

**Affiliations:** 1Laboratory in Phytochemistry Research, Chemical Biological Sciences Department, Universidad de las Américas Puebla, Ex Hacienda Sta. Catarina Mártir S/N, San Andres Cholula 72810, Mexico; luisr.hernandez@udlap.mx; 2Division of Infectious Diseases, Faculty of Medicine, University of British Columbia, Vancouver, BC V6G 3Z6, Canada; hbach@mail.ubc.ca (H.B.); anacecylole@gmail.com (A.C.L.-L.); 3Tecnologico de Monterrey, Escuela de Ingeniería y Ciencias, Campus Guadalajara, Av. Gral. Ramón Corona No 2514, Colonia Nuevo México, Zapopan 45121, Mexico; diegonl@tec.mx (D.E.N.-L.); edgarl@tec.mx (E.R.L.-M.)

**Keywords:** traditional medicine, Crassulaceae, *Kalanchoe fedtschenkoi*, bioactivities, antimicrobial, cytotoxicity, anti-inflammatory properties

## Abstract

In this study, the leaves of *Kalanchoe fedtschenkoi* were consecutively macerated with hexane, chloroform, and methanol. These extracts were used to assess the bioactivities of the plant. The antimicrobial activity was tested against a panel of Gram-positive and -negative pathogenic bacterial and fungal strains using the microdilution method. The cytotoxicity of *K. fedtschenkoi* extracts was investigated using human-derived macrophage THP-1 cells through the MTT assay. Finally, the anti-inflammatory activity of extracts was studied using the same cell line by measuring the secretion of IL-10 and IL-6. The phytoconstituents of hexane and chloroform extracts were evaluated using gas chromatography–mass spectrometry (GC/MS). In addition, high-performance liquid chromatography (HPLC) was used to study the phytochemical content of methanol extract. The total flavonoid content (TFC) of methanol extract is also reported. The chemical composition of *K. fedtschenkoi* extracts was evaluated using Fourier-transform infrared spectroscopy (FTIR). Results revealed that the chloroform extract inhibited the growth of *Pseudomonas aeruginosa* at 150 μg/mL. At the same concentration, methanol extract inhibited the growth of methicillin-resistant *Staphylococcus aureus* (MRSA). Regarding their cytotoxicity, the three extracts were highly cytotoxic against the tested cell line at IC_50_ < 3 μg/mL. In addition, the chloroform extract significantly stimulated the secretion of IL-10 at 50 μg/mL (*p* < 0.01). GC/MS analyses revealed that hexane and chloroform extracts contain fatty acids, sterols, vitamin E, and triterpenes. The HPLC analysis demonstrated that methanol extract was constituted by quercetin and kaempferol derivatives. This is the first report in which the bioactivities and chemical profiles of *K. fedtschenkoi* are assessed for non-polar and polar extracts.

## 1. Introduction

Traditional medicine comprises knowledge, practices, and beliefs that, when integrated, are useful to treat or avert multiple diseases. Given its accessibility and affordability, traditional medicine is utilized as a first line of response against medical emergencies in African countries [1]. In traditional medicine, syrups, decoctions, infusions, and extracts from herbs or medicinal plants are used as antimicrobial, anticancer, anti-inflammatory, antidepressant, or antiaggregant agents [2].

Medicinal plants and their parts are widely used to prepare therapeutic extracts [3]. In traditional medicine, medicinal plants are predominantly utilized to treat diseases, as they are a cost-effective alternative that exerts lower side effects than current treatment modalities [4]. However, over the last decades, there has been an increasing interest in screening the bioactivities of plant extracts against human health concerns such as infectious diseases, inflammatory processes, and distinct types of cancer.

Communicable diseases (CD) are caused by pathogenic microorganisms (e.g., bacteria, fungi, and viruses) or their products [5]. There are various mechanisms by which they can be transmitted, for example, through contact with contaminated objects or blood products, insect bites, and contact with bodily fluids (e.g., saliva). Common examples of communicable diseases include infections caused by hepatitis A and B viruses, Rift Valley fever, influenza, salmonella, and tuberculosis [5,6]. According to the World Health Organization (WHO), infections caused by critical (e.g., *Acinetobacter baumannii* and *Pseudomonas aeruginosa*) and high (e.g., *Staphylococcus aureus*) priority multidrug-resistant bacteria constitute a human health concern, as their incidence has been correlated to the 4.95 million deaths estimated in 2019 [7,8]. However, these numbers are expected to increase in the next decades [9]. Multidrug bacteria infections are challenging to treat due to their limited effectiveness and resistance to current antibiotics. Another major threat to human health is non-communicable diseases (NCDs).

NCDs are also known as chronic diseases. They include a series of clinical conditions characterized by their long duration and gradual progress [10]. Examples of NCDs include cancer, chronic respiratory diseases, diabetes, and cardiovascular diseases. In addition, in view of their clinical features, gastrointestinal diseases, endocrine, neurological, and genetic disorders are also included in this category [11]. Compared to CDs, NCDs represent an increasing concern due to their high mortality rate and impact on the global economy [10]. Epidemiologically, it has been estimated that NCDs account for 71% of all deaths worldwide and predominantly impact low-income and middle-income countries [12].

The Crassulaceae family belongs to the order Saxifragales, or the orpine family or Stonecrop family [13]. Regarding its distribution, the presence of members of the Crassulaceae family is documented in the Mediterranean region, Southern Africa, and the Southwestern United States [14], which includes 35 genera and ∼1410 species [15]. The genus *Kalanchoe* is widely recognized for its ornamental use, mainly attributed to its adaptability to drought, exquisite flowers, easy cultivation, clone growth, and asexual reproduction [16]. In addition, for therapeutic purposes, the biological activities and bioactive compounds from *Kalanchoe* species have been broadly studied.

Over the last years, it has been revealed that *Kalanchoe* species are of great importance in traditional medicine, since they possess different bioactive molecules (e.g., quercetin, afzelin, bryophyllin A, bersaldegenin-3-acetate) that can exert strong antitumor, antimicrobial, anti-inflammatory, antileishmanial, antioxidant, and anti-urolithiasis properties [17,18]. Traditionally, preparations from species of the *Kalanchoe* genus are utilized in different countries such as Brazil, India, and China. In addition, some of them (i.e., *K. pinnata*) belong to the list of medicinal plants to be used in national public health systems, such as the Sistema Único de Saúde (SUS) [19].

The importance of *Kalanchoe* species relies on their capacity to exert different biological activities. Therefore, they have been proposed to treat rheumatic disorders, abscesses, wounds, and burns [20]. Comparably, it has been reported that extracts from their leaves can execute hepatoprotective, hypocholesterolemic, nephroprotective, and nematicide activities [21]. In addition, species from the genus are used in traditional medicine to induce smooth muscle relaxation [22] and prevent premature labor [23]. Moreover, the increasing interest in expanding the therapeutic knowledge about *Kalanchoe* species has resulted in their use as transgenic plants to produce peptides (cecropin P1) with fungicide and wound-healing activity in Wistar rats [24].

*K. fedtschenkoi*, also known as *Bryophyllum fedtschenkoi*, is a native species from Madagascar that has been poorly studied. For instance, only one study reported that the ethanol extract from this specimen exhibited antibacterial and cytotoxic properties against the group of ESKAPE (*Enterococcus faecium*, *S. aureus*, *Klebsiella pneumoniae*, *A. baumannii*, *P. aeruginosa*, and *Enterobacter cloacae*) pathogens and human keratinocytes, respectively [25]. Furthermore, the same study demonstrated that ethanol extract contained different bioactive nature compounds such as quercetin and caffeic acid [25]. On the other hand, recent studies have revealed that aqueous extracts from its leaves can exert antioxidant activity, since they contain distinct flavonoids such as quercetin di-*O*-hexoside, methylquercetin-*O*-hexoside-*O*-deoxyhexoside, kaempferol 3-*O*-glucopyranoside 7-*O*-rhamnopyranoside, and kaempferol-*O*-hexoside-*O*-deoxyhexoside-*O*-pentoside, among others [26].

Continuing with our research program of studying the bioactivities of traditional medicinal plants, we examined the antimicrobial and cytotoxic activities of hexane, chloroform, and methanol extracts from *K. fedtschenkoi* leaves. The antimicrobial activity of *K. fedtschenkoi* extracts was evaluated against a panel of pathogenic Gram-positive and -negative bacteria and yeast strains. In addition, these extracts’ cytotoxicity and inflammatory response were studied using a human-derived monocyte cell line as an ex vivo model.

## 2. Results and Discussion

### 2.1. GC/MS Analysis

Hexane and chloroform extracts contained fatty acids, vitamins, sterols, and triterpenoids commonly found in the *Kalanchoe* genus (Table 1). For example, among *Kalanchoe* species, fatty acids such as heptacosane and stearic acid have been unveiled from *K. beharensis* and *K. pinnata*, respectively [27,28]. Similarly, *K. pinnata* leaves have been reported as abundant sources of sterols such as stigmasterol [29] and triterpenes such as friedelin [30]. These compounds and diterpenes, such as phytol, have been identified in *K. tomentosa* extracts [31]. On the other hand, fat-soluble phenolic molecules such as vitamin E, also known as tocopherol, have been documented among *K. daigremontiana* [32] and *K. crenata* extracts [33]. The chromatograms of hexane and chloroform extract are presented as Appendix A. Given the polarity of methanol extract, its phytochemical content was studied using HPLC, and it was compared with the literature reporting on polar extracts from *K. fedtschenkoi*.

### 2.2. HPLC Analysis

HPLC is a proper technique in which analytes dissolved in a mobile phase are pumped through a stationary phase. Depending upon the chemical features of the sample, solvent, and stationary phase, analytes exhibit different retention times (Rt). Here, we used HPLC to study the chemical composition of methanol extract from *K. fedtschenkoi*. As presented in the Appendix A, the chromatogram of methanol extract exhibits characteristic peaks corresponding to flavonoids identified in other reports in which *K. fedtschenkoi* has also been studied [25,26]. In accordance with their results, we show that methanol extract also contains quercetin di-*O*-hexoside (Rt: 14.98), methylquercetin-*O*-hexoside-*O*-deoxyhexoside (Rt: 17.37), kaempferol *O*-hexoside-di-*O*-deoxyhexoside (Rt: 19.34 min), and kaempferol *O*-hexoside-di-*O*-deoxyhexoside (Rt: 20.13 min). On the other hand, the same chromatogram also presents a small peak at 38.54 min and a sharp peak at 47.23 min, which, comparably to HPLC analyses of extracts from *K. brasiliensis*, might suggest the presence of patuletin-*O*-deoxy-hexoside-*O*-acetyl-deoxy-hexoside [34]. To estimate the amount of flavonoids in the methanol extract, we performed the TFC assay.

### 2.3. TFC of Methanol Extract from K. fedtschenkoi

The TFC assay is a widely performed colorimetric method required to assess the presence of flavonoids in plant extracts. This technique is based on aluminum chloride (AlCl_3_)’s capacity to form complexes with hydroxyl and carbonyl groups from various flavonoids.

Flavonoids constitute a broad category of secondary metabolites. Structurally, flavonoids are formed by two benzene rings (A and B) joined by a three-carbon-based pyran ring (C). According to their substitution pattern and the number of functional groups, flavonoids are categorized into anthocyanidins, flavonols, flavones, and isoflavones. It is known that *Kalanchoe* species can contain multiple flavonoids such as quercetin (Qu), kaempferol, myricetin, luteolin, eupafolin, or their derivatives [35]. These compounds have been reported among several species, for example, *K. pinnata*, *K. gracilis*, *K. blossfeldiana*, *K. tomentosa*, and *K. pathulate* [19,20,36].

A calibration curve was constructed considering various concentrations of Qu to estimate the TFC of methanol extract from *K. fedtschenkoi*. To perform the TFC assay, bioactive nature products such as catechin and rutin are commonly used. However, Qu is also preferred, as it is a flavonol that reacts with AlCl_3_ due to its keto group at C4 and hydroxyl groups at C3 or C5. Therefore, using the regression equation (*y* = 0.0007*x* + 0.3518, *R*^2^ = 0.9995) presented in the Appendix A, the TFC of this extract was estimated and represented in milligrams of quercetin equivalents per gram of the plant extract (mg Qu/g). In this regard, *y* was considered as the absorbance of the test sample, whereas *x* was appraised as the concentration from the calibration curve. Following our calculations, the TFC of methanol extract is 384.54 ± 2.25 mg Qu/g. This result can be comparable to the TFC of methanol extracts prepared from other *Kalanchoe* species, such as *K. pinnata* (106 mg Qu/g) and *K. integra* (178 mg Qu/g) [37].

### 2.4. Antimicrobial Activity

In this study, the chloroform and methanol extract from *K. fedtschenkoi* exhibited weak antimicrobial activity against the panel tested (see Table 2). For instance, treatment with 150 μg/mL of chloroform extract only inhibited the growth of *P. aeruginosa*. At the same concentration, methanol extract inhibited the growth of MRSA. No inhibition of yeast strains was recorded.

Medicinal plants can exhibit antimicrobial properties against pathogenic bacteria, fungi, protozoa, and viruses through bioactive secondary metabolites such as alkaloids, flavonoids, terpenes, and polysaccharides. Generally, extracts from medicinal plants or herbs can inhibit the growth of pathogenic bacteria and fungi by damaging cell membranes or walls, interfering with protein synthesis, and increasing intracellular osmotic pressure [38]. These mechanisms are due to the phytochemical content of plant extracts; for example, given the existence of hydroxyl groups and delocalized electrons among polyphenols’ architecture, they can increase the permeability of the bacterial membrane, alter its potential, and cause structural changes [39]. In contrast, the acyl chains, numerous hydroxyl groups, and glycosylated moieties of flavonoids enable their capacity to reduce nucleic acid synthesis, disrupt energy metabolisms, and suppress cytoplasmic bacterial membrane functionality [40].

Among *Kalanchoe* species, many antibacterial and antifungal compounds have been identified over the last decades. For example, *K. pinnata* and *K. daigremontiana* are well-known medicinal plants that contain flavonols (e.g., quercetin and kaempferol), flavones (e.g., luteolin), and bufadienolides [41] that can reduce the formation of biofilms, inhibit the growth of pathogenic bacteria strains, decrease protein synthesis, or inhibit the expression of genes related to antimicrobial resistance [20,42,43,44]. According to published reports [45], these mechanisms of action might be related to the glycosyl derivatives of kaempferol identified on the methanol extract of *K. fedtschenkoi*. In the same regard, extracts from other species, such as *K. blossfeldiana*, contain palmitic acid, gallic acid, methyl gallate, and carbohydrates that can be correlated to their antimicrobial properties [46].

Among multidrug-resistant bacteria, *P. aeruginosa* is a challenging pathogen in human health care. It can cause acute or chronic infections in patients diagnosed with cystic fibrosis, cancer, and coronavirus disease-19 (COVID-19) [47]. Only one study has reported the inhibition of *P. aeruginosa*, using ethanol extracts from *K. fedtschenkoi* at concentrations ranging from IC_50_ 128 to 256 μg/mL; similar concentrations were reported against *A. baumannii* and *S. aureus* [25]. These findings can be attributed to differences in harvesting places, climate, soil characteristics, extract polarity, implemented methodology, and strain culture conditions.

MRSA is a global human health threat characterized by its prevalence in the community, ease of spread, and capacity to cause endocarditis, bacteremia, osteomyelitis, pneumonia, and purulent infections [48]. These results can be compared with the activity of hydroethanolic extracts from *K. brasiliensis* that have inhibited the growth of MRSA strains at MIC > 5000 μg/mL. The antibacterials eupafolin and patuletin or their glycosylated derivatives can explain this activity [49].

### 2.5. Cytotoxicity Activity

The evaluation of the toxicity of plant extracts or bioactive nature products is necessary to determine their possible application in the development of pharmaceutical formulations or use against other diseases, such as cancer [50]. In this study, we assessed the cytotoxicity using the MTT ((3-(4,5-dimethylthiazol-2-yl)-2,5-diphenyltetrazolium bromide) assay [51]. Results demonstrated that the three extracts exhibited a significant cytotoxic effect (*p* < 0.0001) at the tested concentrations (Figure 1). In this regard, it should be noted that even though treatment with hexane and chloroform extracts was cytotoxic to THP-1 cells, treatment with the methanol extract exhibited the highest cytotoxicity towards the tested cell line (*p* < 0.0001).

It is known that medicinal plants have represented an exceptional source of cytotoxic molecules with potential use against various cancers such as lung, breast, and prostate cancer. This is of broad importance in developing countries such as Mexico, since approximately 30% of Mexican patients utilize preparations from medicinal plants as a preventive, complementary, and cost-effective approach during cancer therapy [52]. Furthermore, the structural diversity of isolated molecules from medicinal plants is exploited in developed countries to develop drugs against infectious and neoplastic diseases [53,54].

According to the National Cancer Institute (NCI) of the U.S., the cytotoxicity of molecules can be considered high (IC_50_ ≤ 20 μg/mL), moderate (IC_50_ 21–200 μg/mL), or weak (IC_50_ 201–500 μg/mL) [55,56]. We found that the hexane, chloroform, and methanol extracts were highly cytotoxic against the tested cell line, as they presented IC_50_ values of 2.090, 1.918, and 1.722 μg/mL, respectively. This result can be attributed to the extracts’ polarity and their phytoconstituents.

For instance, the cytotoxicity of the hexane extract might be due to the presence of friedelin, which has been reported to inhibit the proliferation of breast cancer cells [57]. In the same regard, the cytotoxicity of the chloroform extract can be attributed to the presence of stigmasterol and tocopherol. Currently, the former is recognized as a promising cytotoxic agent for cancer therapy [58], whereas the latter can exert distinct cytotoxic effects on cell lines such as bovine endothelial cells, mouse macrophages, and human hepatocytes [59]. On the other hand, the cytotoxicity of the methanol extract might be due to synergism between the derivatives of quercetin and kaempferol, which have been reported to induce the death of different cell lines such as MDA-MB-231 and MCF-7 [60].

Non-polar solvents such as hexane and chloroform are frequently used to extract bioactive compounds such as terpenoids, fats, and oils. In contrast, polar solvents, such as methanol, are preferably used, as they are accessible, nontoxic at low concentrations, and can extract high polar compounds with strong therapeutic properties (e.g., anthocyanins, polyphenols, and flavonoids) such as antimicrobial and anticancer qualities [4,61]. Interestingly, the cytotoxicity of *Kalanchoe* species has been widely reported for polar extracts in the scientific literature. For instance, treatment with less than <40 μg/mL of *K. crenata* methanol extract was cytotoxic towards breast adenocarcinoma (MCF-7), hepatocarcinoma (HepG2), colorectal adenocarcinoma (DLD-1), and human non-small-cell lung cancer (A549) cell lines [62].

In another study, comparable effects have been documented for ethanol extracts prepared from the leaves of *K. millotii* and *K. nyikae*, which were cytotoxic against human acute lymphoblastic leukemia T (J45) and human T (H9) cell lines at IC_50_ values of 503.5 and 560.5, and 846.1 and 507.6 μg/mL, respectively [63]. Against other types of cancer, such as the human ovarian cancer SKOV-3 cells, a water extract from the leaves of *K. daigremontiana* arrested the cell cycle and induced mitochondrial membrane depolarization. It was cytotoxic against the cells at 5 to 200 μg/mL [32]. Comparably to these reports, our work demonstrated that the *K. fedtschenkoi* extracts’ cytotoxicity increases according to the polarity of the solvents. For instance, treatment with the methanol extract was highly cytotoxic against THP-1 cells at 1.722 μg/mL, whereas treatment with hexane extract decreased cell viability at 2.090 μg/mL. In another study, the cytotoxicity of the ethanol extract from *K. fedtschenkoi* was evaluated towards human keratinocytes cells (HaCAts); this study revealed that ethanol extract decreased cell viability at LD_50_ > 250 μg/mL [25]. However, other differences between our results and other reports can be attributed to the studied species, tested concentrations, extracts’ polarity, and evaluated cell lines.

### 2.6. Anti-Inflammatory Activity

Inflammation occurs when hazardous stimuli, such as microorganisms, toxic compounds, or damaged cells, activate immune cells [64]. Multidrug-resistant bacteria can promote inflammation processes through different mechanisms. Inflammatory responses caused by pathogenic bacteria arise from the interaction between pathogen-associated molecular patterns (PAMPs) and pattern-recognition receptors (PRRs), which are structures expressed among immune and non-immune cells [64]. PRRs recognize distinct ligands from bacterial architecture, such as flagellin, peptidoglycans, genetic material, and lipoteichoic acid [65]. The interaction between PAMPs and PRRs results in complex intracellular signaling cascades devoted to recruiting inflammatory molecules necessary to retain the progression of infection and inflammation and to initiate tissue repair [66,67].

In this study, we show that treatment with the *K. fedtschenkoi* chloroform extract significantly stimulated the secretion of the anti-inflammatory cytokine IL-10 at 50 μg/mL (*p* < 0.01). Still, it did not reduce the secretion of the pro-inflammatory cytokine IL-6 at the tested concentration (Figure 2). The anti-inflammatory activity of the chloroform extract can be attributed to the presence of stigmasterol. This widely recognized phytosterol can suppress the production of IL-6 and other pro-inflammatory cytokines such as tumor necrosis factor-α (TNF-α) [68,69].

On the other hand, treatment with hexane and methanol extract enhanced the secretion of IL-6, which can be attributed to the cytotoxicity of these extracts. Several models have been proposed to evaluate the anti-inflammatory properties of *Kalanchoe* species. For example, it has been reported that treatment with *K. pinnata* hydroethanolic extract reduced pro-inflammatory responses among rodent models by downregulating the expression of mediators involved in inflammatory processes such as Toll-like receptors and nuclear factor kappa B (NFκB) [70]. Comparably, topical formulations prepared from *K. pinnata* and *K. brasiliensis* aqueous extracts decreased the production of TNF-α and IL-1β while enhancing the levels of IL-10 among edematogenic Swiss mice models [71].

### 2.7. FTIR Analysis

FTIR spectroscopy is based on the absorption of infrared light by organic and inorganic analytes. Each sample exhibits a distinctive spectrum fingerprint in this technique that can be recognized and differentiated from other molecules [72]. For plant extracts, FTIR can be used to study their chemical composition in solid and liquid samples.

Figure 3 depicts *K. fedtschenkoi* extracts exhibiting similar bands within 2916 and 2849 cm^−1^, corresponding to the asymmetrical and symmetrical stretching of the C-H bonds from hydrocarbon chains. Comparably, hexane and methanol extracts present a broad band at 3298 cm^−1^, related to O-H bond stretching, usually associated with phenolic compounds. This finding might suggest the presence of compounds reported among *K. fedtschenkoi* extracts, such as quercetin. Among bioactive nature products, quercetin is an abundant flavonoid in plants, fruits, and vegetables, exhibiting different peaks in the infrared region. It has been reported that flavonoids such as quercetin exhibit distinctive bands from 1610 to 1510 cm^−1^, attributed to C=C bonds from its aromatic ring. In addition, it is documented that quercetin presents a series of bands from 1260 to 1160 cm^−1^, which is related to the C-O bond stretch aryl ether ring of its structure [73]. Interestingly, FTIR spectra of *K. fedtschenkoi* extracts present related peaks from 1600 to 1500 cm^−1^, which can respond to bending vibrations of C=O bonds, possibly from esters, ketones, and carboxylic acids [74]. These findings are in accord with FTIR analysis of the chemical composition of extracts from other plants, including *K. fedtschenkoi* [75,76].

## 3. Materials and Methods

### 3.1. Plant Material and Extract Preparation

Aerial parts of *K. fedtschenkoi* were collected in Texcoco, Mexico (19.491100418879938, −98.88525021669616). Specimens were identified by the biologist Lilián López-Chávez at the herbarium of Universidad Autónoma de Chapingo (Carr. Federal México-Texcoco, 56230, Texcoco, Estado de México) and deposited with the voucher number 36205. The collected fresh plants were dried in a ventilated and dark place at room temperature for one week. For extract preparation, 500 g of dried leaves were finely powdered utilizing a mechanical blender before being initially macerated with hexane, then with chloroform, and finally, with methanol for three days (1.5 L each). After that, the mixture was filtrated, and the solvent was evaporated to dryness under reduced pressure using a Heidolph Laborota 4000 efficient rotary evaporator (Schwabach, Germany). The extracts were preserved under refrigeration until further use.

### 3.2. GC/MS Analysis

The phytoconstituents of *K. fedtschenkoi* extracts were identified using a Varian CP-3800 gas chromatograph coupled to a Varian 1200 quadrupole mass spectrometer [77]. Samples were injected into a Factor Four (VF-5 ms, 30 m × 0.25 mm, 0.25 μm thickness) capillary column. Helium was used as the carrier gas at a 1 mL/min flow rate. The separation was carried out by injecting 1 μL of the sample (1%) into the column at the following gradient temperature: 60 °C for 2 min, 120 °C for 16 min, 30 °C/min up to 150 °C for 15 min, 20 °C/min up to 180 °C for 15 min, 30 °C/min up to 200 °C for 10 min, 20 °C/min up to 220 °C for 15 min, 5 °C/min up to 280 °C for 20 min, and 5 °C/min up to 300 °C for 20 min. Individual components from the extracts were identified based on comparing their retention times and fragmentation patterns to the National Institute of Standards and Technology Mass Spectral (NIST-MS) database. The total area of the peaks assessed their relative percentage.

### 3.3. HPLC Analysis

The phytochemical content of methanol extract was studied considering reported protocols with minor modifications [26]. Briefly, HPLC analysis was performed on an Agilent Technologies 1200 series equipped with a diode array detector (DAD) utilizing reagents of HPLC grade. An RP-18 Zorbax 150 mm × 4.6 mm, 3.5 μm column was employed as a stationary phase. On the other hand, water acidified with 0.1% formic acid (A) and acetonitrile (B) were used as the mobile phase. To prepare the sample, 2 mg of extract was diluted in water/acetonitrile (5:1). To carry out analysis, 10 μL of sample was run at the following gradient: 0–20 min (0–20% B), 20–40 min (20–22% B), 40–43 min (22–30% B), 43–45 min (30–100%B), 45–50 min (100% B). Absorbance was monitored at 254 and 365 nm. The analysis was performed in triplicate.

### 3.4. TFC Analysis

The TFC of *K. fedtschenkoi* methanol extract was determined as published [78]. Shortly, 100 μL of methanol extract (1 mg/mL) mixed with 100 μL of 2% aluminum chloride (AlCl_3_) was incubated for 10 min. Then, using a Cary 60 UV-Vis spectrophotometer (Agilent Technologies, Santa Clara, CA, USA), absorbance was measured at 420 nm in 1 cm quartz cuvettes. A standard curve was obtained using distinct concentrations (100 to 500 μg/mL) of a standard solution of quercetin (Qu). The TFC of methanol extract was estimated as a percentage of total quercetin equivalents per gram of extract (mg Qu/g). The experiment was performed in triplicate.

### 3.5. Strains and Culture Media

The antimicrobial activity of *K. fedtschenkoi* extracts was studied against a panel of pathogenic Gram-positive and -negative bacteria and yeast strains. Gram-positive bacteria included *Listeria monocytogenes* (ATCC BAA-679), methicillin-resistant *Staphylococcus aureus* (MRSA) (ATCC 700698), and *Staphylococcus aureus* (ATCC 25923). On the other hand, Gram-negative bacteria included *Acinetobacter baumannii* (ATCC BAA-747), *Escherichia coli* (ATCC 25922), and *Pseudomonas aeruginosa* (ATCC 14210) strains. Clinical isolates of *A. baumannii* and *P. aeruginosa* were also tested in this study [79]. In contrast, fungal strains included *Candida albicans* (ATCC 10231) and *Cryptococcus neoformans* (kindly provided by Dr. Karen Bartlet, University of British Columbia, Vancouver, BC, Canada). Bacteria were cultured in Mueller–Hinton broth (B&D) at 37 °C, and fungal strains were cultured in Sabouraud broth (B&D) at 28 °C. In both cases, a shaker was used.

### 3.6. Microdilution Assay

The antimicrobial assay was conducted as in previously published protocols [80]. Shortly, in a 96-well plate, a microdilution assay was performed to determine the minimum inhibitory concentration (MIC). In 100 μL/well of Mueller–Hinton or Sabouraud broth, the following concentrations of *K. fedtschenkoi* extracts were tested against different microbial inocula: 50, 100, 150, and 200 μg/mL. The inoculum was prepared to have a final optical density of 0.05 at 600 nm. Bacteria treated with amikacin and gentamicin were used as positive controls, whereas untreated cells and DMSO were used as negative controls. For fungi, amphotericin and terbinafine were used as positive controls. All experiments were executed in triplicate.

### 3.7. Cytotoxicity Assay

The cytotoxicity of 50, 100, 150, and 200 μg/mL *K. fedtschenkoi* extracts was evaluated using human-derived THP-1 monocytic cells (ATCC TIB-202), following published protocols [81]. Using RPMI 1640 (Hyclone, GE Healthcare, Logan, UT, USA) supplemented with 5% fetal bovine serum (FCS) (Hyclone) and 2 mmol L^−1^ L-glutamine (Stem Cell Technologies, Vancouver, BC, Canada), THP-1 cells were cultured. To differentiate these cells, 20 ng/mL phorbol 12-myristate 13-acetate (PMA) was used. Then, cells were dispensed in a 96-well plate with a final volume of 100 μL at a final concentration of 1 × 10^5^ cells per well. The plate was incubated at 37 °C with 5% CO_2_ for 24 h. The next day, the medium was removed and replaced with fresh medium, and *K. fedtschenkoi* extracts at concentrations mentioned above. Again, the plate was incubated at 37 °C supplemented with 5% CO_2_ for 24 h. Untreated cells and DMSO served as negative controls. Cells treated with 2% Tween-20 were used as a positive control. According to the same protocol [81], the cytotoxicity of *K. fedtschenkoi* extracts was known by the 3-(4,5-dimethylthiazol-2-yl)-2,5-diphenyltetrazolium bromide (MTT) assay. The next day, 2 h prior to the end of the incubation period, 25 μL of a working solution of MTT (5 mg mL^−1^) was added to the cells, and they were incubated for a further 4 h. To dissolve formazan, 100 μL of extraction buffer was added per well; the plate was incubated overnight at 37 °C. In a plate reader, readings were performed at 570 nm. The half-maximal inhibitory concentration (IC_50_) was estimated by plotting the log concentrations of the extracts against the percentage of damaged cells. All experiments were performed in triplicate.

### 3.8. Anti-Inflammatory Assay

To study the anti-inflammatory activity of *K. fedtschenkoi* extracts, published protocols were followed [81]. In brief, THP-1 cells differentiated with PMA were used in a 96-well plate at a final concentration of 1 × 10^5^ cells per well. Given the results obtained during the cytotoxicity assay, *K. fedtschenkoi* extracts were assayed at a final concentration of 50 μg/mL. Cells treated with 100 ng/mL of lipopolysaccharide (LPS) from *E. coli* (Sigma-Aldrich, St. Louis, MO, USA) were used as a positive control. Contrarily, cells treated with DMSO were used as a negative control. The final concentration of DMSO per well was always ≤1%. The measurement of the pro-inflammatory cytokine IL-6 and the anti-inflammatory cytokine IL-10 was executed with commercial kits (B&D) following instructions from the manufacturer. Readings were recorded using a plate reader at 450 nm. All experiments were carried out in triplicate.

### 3.9. FTIR Analysis

The chemical composition of *K. fedtschenkoi* extracts was studied using a Cary 630 Fourier-transform infrared (FTIR) spectrometer (Agilent Technologies, Santa Clara, CA, USA). Briefly, the detection diamond was cleaned with 10 μL of ethanol solution (100% *v*/*v*) and allowed to air dry before analyses. Background spectra were recorded without samples at room temperature. For sample analysis, 20 mg of each extract was used per reading, and the crystal was cleaned after each measurement. Measurements were recorded within the 4000 to 400 cm^−1^ wavenumber region. All readings were recorded in triplicate.

### 3.10. Statistical Analysis

Data from the quantitative viability analysis were subjected to two-way analysis of variance (ANOVA), followed by a Tukey’s mean separation test, to determine the relationship between each treatment using OriginPro 2023 data processing software (OriginLab, Northampton, MA, USA).

## 4. Conclusions

This study demonstrated that *K. fedtschenkoi* chloroform and methanol extract exhibit antibacterial activity against multidrug-resistant bacteria such as *P. aeruginosa* and MRSA.

During the cytotoxicity assay, the three extracts were highly cytotoxic against THP-1 cells with IC_50_ < 3 μg/mL, which suggests their potential use against cancer in in vitro or in vivo models. Furthermore, statistical analysis revealed that methanol extract exhibited higher cytotoxicity (*p* < 0.0001) against the tested cell line than hexane and chloroform extract. On the other hand, treatment with *K. fedtschenkoi* chloroform extract promoted the secretion of the anti-inflammatory cytokine IL-10.

Regarding their phytochemical content, this work demonstrated that hexane and chloroform extracts are mainly comprised of fatty acids, sterols, and triterpenes. In contrast, the derivatives of flavonoids such as quercetin and kaempferol are mainly in the methanol extract. In addition, we determined that methanol extract is abundant in flavonoids, as it presented a TFC of 384.54 ± 2.25 mg Qu/g. Using FTIR spectroscopy, we also demonstrated that extracts from the leaves of *K. fedtschenkoi* contain several functional groups such as C-H, C-O, C=O, C=C, and OH groups that might be related to compounds identified during the GC/MS or HPLC analyses performed in this work.

To the best of our knowledge, this is the first report that details the antimicrobial, cytotoxic, and anti-inflammatory properties of non-polar and polar extracts from *K. fedtschenkoi*. In addition, this study expands upon the knowledge on the genus *Kalanchoe* and assesses the importance of continuing to explore its therapeutic potential.

## Figures and Tables

**Figure 1 plants-12-01943-f001:**
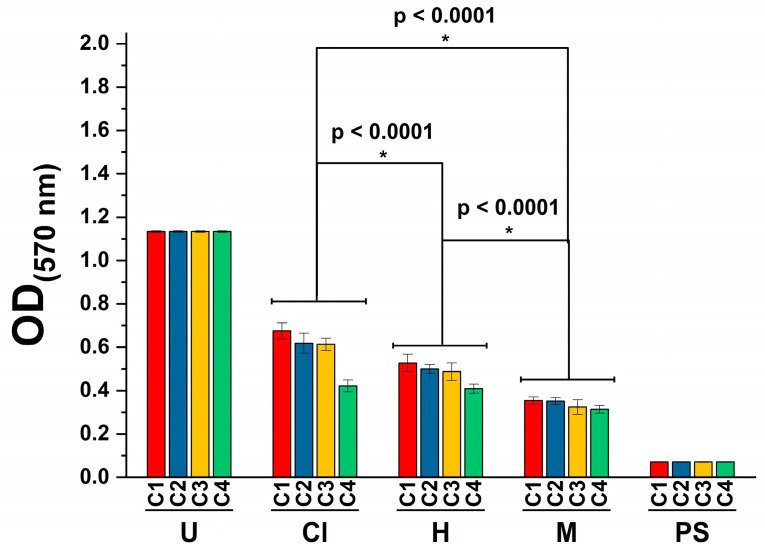
Viability of human-derived THP-1 cells against treatment with 50 (C1), 100 (C2), 150 (C3), and 200 (C4) μg/mL of *K. fedtschenkoi* using U, untreated cells, chloroform (Cl), hexane (H), and methanol (M) extracts using the MTT assay. PS, positive control (2% Tween-20); OD, optical density. Shown is the mean ± S.D. of three independent experiments. * Represents *p*-values significantly below to <0.0001 evaluated with Tukey’s test.

**Figure 2 plants-12-01943-f002:**
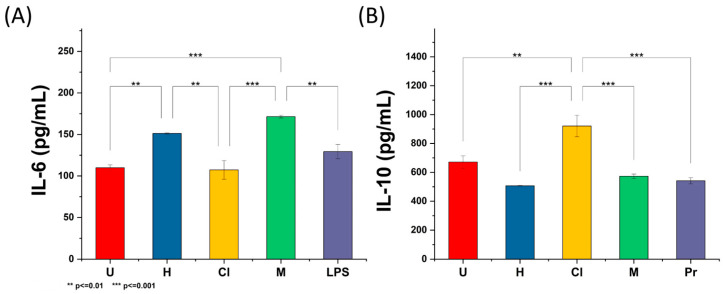
Immunological response of *K. fedtschenkoi* hexane (H), chloroform (Cl), and methanol (M) extracts on human-derived THP-1 cells using ELISA for (**A**) IL-6 and (**B**) IL-10. LPS, lipopolysaccharide (positive control for inflammation), untreated cells (U), and prednisone (Pr, anti-inflammatory, positive control). Shown is the mean ± S.D. of three independent experiments. ** Represents *p*-values significantly below <0.01 evaluated with Tukey’s test. *** Represents *p*-values significantly below <0.001 evaluated with Tukey’s test.

**Figure 3 plants-12-01943-f003:**
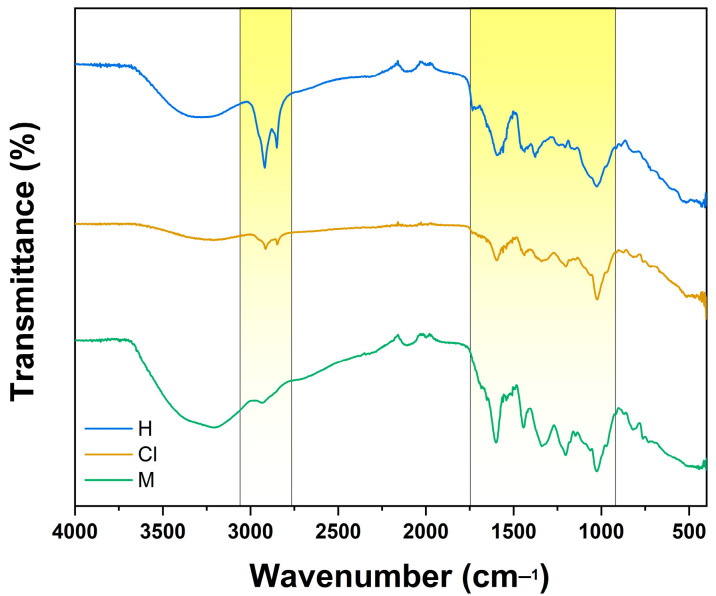
FTIR analysis of *K. fedtschenkoi* hexane (blue line), chloroform (yellow line), and methanol (green line).

**Table 1 plants-12-01943-t001:** Chemical composition of hexane and chloroform extracts from *K. fedtschenkoi*.

Extract	Match	R match	Rt (min)	%	Name
	749	776	45.11	0.59	n-hexadecanoic acid
	888	902	54.86	1.74	Phytol
	735	750	58.22	0.58	Stearic acid
	885	916	94.96	1.23	Squalene
Hexane	878	883	103.47	3.83	δ-Tocopherol
	770	809	109.55	0.84	ε-Tocopherol
	883	887	116.01	4.35	α-Tocopherol
	767	771	125.10	0.88	Stigmasterol
	919	925	128.74	15.34	Heptacosane
	813	884	131.90	1.44	Simiarenol
	854	856	138.60	1.90	Friedelin
	759	804	148.48	0.62	Octadecanal
	704	723	151.54	0.305	2-Hexadecanol
	756	785	37.82	4.34	Phytol
	726	756	45.09	1.86	Hexadecanoic acid
	757	772	58.28	2.97	Stearic acid
Chloroform	678	750	95.54	0.44	1-Hexadecanol
	680	683	97.30	0.24	1-Pentatriacontanol
	853	862	103.79	4.85	δ-Tocopherol
	747	850	109.43	5.48	1-Docosene
	812	850	110.75	3.83	ε-Tocopherol
	856	869	115.85	8.13	α-Tocopherol
	724	730	122.14	4.48	β-Stigmasterol
	895	913	128.09	24.22	Heptacosane
	750	836	131.18	2.34	β-Simiarenol
	748	799	136.77	0.25	Octadecanal
	742	861	148.59	0.62	Hexadecanal

Abbreviations: Rt, retention time; min, minutes.

**Table 2 plants-12-01943-t002:** Antimicrobial activity of *K. fedtschenkoi* extracts expressed as the minimal inhibitory concentration (μg/mL).

Extract		Bacteria							Fungi	
	MRSA	SA	AB	PA	EC	LM	ABc	PAc	CA	CN
Hexane	R	R	R	R	R	R	R	R	R	R
Chloroform	R	R	R	150	R	R	R	R	R	R
Methanol	150	R	R	R	R	R	R	R	R	R

Abbreviations: MRSA, methicillin-resistant *Staphylococcus aureus*; SA, *Staphylococcus aureus*; AB, *Acinetobacter baumannii*, PA, *Pseudomonas aeruginosa*; EC, *Escherichia coli*; LM, *Listeria monocytogenes*; ABc, *Acinetobacter baumannii* clinical isolate; PAc, *Pseudomonas aeruginosa* clinical isolate; CA, *Candida albicans*; CN, *Cryptococcus neoformans*; R, resistant.

## Data Availability

The information generated in this study can be consulted with authors for correspondence from this work.

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
