# Peer review of "Biological Activities and Chemical Profiles of Kalanchoe fedtschenkoi Extracts"

_plants, 2023, doi:10.3390/plants12101943_

Round 1

Reviewer 1 Report

Congratulations!

Reviewer 2 Report

The MS presents an interesting reseach topic and provides novel data on biological activity of extracts originating from Kalanchoe fedtschenkoi. However, it should be emphasized that the performed experiments had a very general character and no mechanistic studies have been presented. The study design is based on simple in vitro tests. Furthermore, the study is completely lacking a detailed phytochemical analyses of the examined extracts. The authors have provided only some general information derived from the FTIR spectroscopy analyses. No data on qualitative and/or quantitative phytochemical analyses have been given. The authors should provided phytochemical data, including (at least) the total phenolic content and some information on main groups of plant metabolites that are present in the examined plant extracts.

Reviewer 3 Report

Article

Biological Activities of Kalanchoe fedtschenkoi extracts

A brief summary

The work appears to be interesting and quite valuable. The research is reasonably well designed, executed and documented. The presentation and discussion of the results may raise some concerns. The style of presentation is roughly correct, although both the discussion and the conclusion are a bit overdone. The authors have overemphasised the background and expectations of the preparations possible to obtain from plants compared to the actual presentation of their own results. The text also needs some minor linguistic correction.  

Broad comments

    1. The leaves of this plant are quite fleshy. Is drying by placing only “in a ventilated and dark place” (line 301) a suitable method of preparing the raw herbal material? What about the temperature and drying time?

    2. One may have reservations about the statistical analysis. If a post-hoc analysis was carried out using the Tukey test, why are the results obtained using it not presented in the graphs? It cannot be concluded from Figure 1 that "Only the chloroform extract's 200 μg/mL concentration 171 showed a significant difference in this treatment" (lines 171-172). Figure 2 is also illegibly designed. 

    3. A qualitative and quantitative analysis of the extracts tested is clearly lacking. FTIR spectroscopy alone, however valuable, is not enough in such studies.

Round 2

Reviewer 2 Report

The manuscript has been improved.

Reviewer 3 Report

The authors took great adherence to the corrections, which completely transformed the article. It is now suitable for publication.